# Towards Characterizing the First-order Query Complexity of Learning (Approximate) Nash Equilibria in Zero-sum Matrix Games

**Hedi Hadiji**
Laboratoire des signaux et systèmes
Univ. Paris-Saclay, CNRS, CentraleSupélec

**Sarah Sachs, Tim van Even**
Korteweg-de Vries Institute for Mathematics
University of Amsterdam

**Wouter Koolen**
Centrum Wiskunde & Informatica and University of Twente

## Abstract

In the first-order query model for zero-sum $K \times K$ matrix games, players observe the expected pay-offs for all their possible actions under the randomized action played by their opponent. This classical model has received renewed interest after the discovery by Rakhlin and Sridharan that $\varepsilon$-approximate Nash equilibria can be computed efficiently from $O(\ln K/\varepsilon)$ instead of $O(\ln K/\varepsilon^2)$ queries. Surprisingly, the optimal number of such queries, as a function of both $\varepsilon$ and $K$, is not known. We make progress on this question on two fronts. First, we fully characterise the query complexity of learning exact equilibria ($\varepsilon = 0$), by showing that they require a number of queries that is linear in $K$, which means that it is essentially as hard as querying the whole matrix, which can also be done with $K$ queries. Second, for $\varepsilon > 0$, the current query complexity upper bound stands at $O(\min(\ln(K)/\varepsilon, K))$. We argue that, unfortunately, obtaining a matching lower bound is not possible with existing techniques: we prove that no lower bound can be derived by constructing hard matrices whose entries take values in a known countable set, because such matrices can be fully identified by a single query. This rules out, for instance, reducing to an optimization problem over the hypercube by encoding it as a binary payoff matrix. We then introduce a new technique for lower bounds, which allows us to obtain lower bounds of order $\tilde{\Omega}(\log(\frac{1}{K\varepsilon}))$ for any $\varepsilon \leqslant 1/(cK^4)$, where $c$ is a constant independent of $K$. We further discuss possible future directions to improve on our techniques in order to close the gap with the upper bounds.

## 1 Introduction

Computing the saddle point

$$\min_{x \in \mathcal{X}} \max_{y \in \mathcal{Y}} f(x, y)$$

for convex-concave functions $f : \mathcal{X} \times \mathcal{Y} \to \mathbb{R}$ is of general interest throughout optimization, economics and machine learning. We study the computation of an approximate saddle point $(x_\star, y_\star)$, satisfying $\max_{y \in \mathcal{Y}} f(x_\star, y) - \min_{x \in \mathcal{X}} f(x, y_\star) \leqslant 2\varepsilon$ for some given $\varepsilon \geqslant 0$. Starting with an unknown $f$ from a known class $\mathcal{F}$, we consider sequential learning in the first-order feedback model, where the learner gets to observe gradients of the objective. Formally, each query $(x, y)$ results in feedback $(\nabla_x f(x, y), \nabla_y f(x, y))$. We are interested in the following question:

*How many first-order queries are necessary and sufficient for a sequential learner to output an approximate saddle point for any $f \in \mathcal{F}$?*

37th Conference on Neural Information Processing Systems (NeurIPS 2023).

Characterizing the query complexity of learning saddle points is of theoretical interest for understanding the hardness of computing equilibria, and for certifying the optimality of upper bounds.

In this work, we restrict our attention to the special case of zero-sum matrix games, where $\mathcal{X}$ and $\mathcal{Y}$ are finite-dimensional probability simplices and $f$ is bilinear. For this canonical setting, the optimal query complexity is, surprisingly, unresolved. Indeed, computation algorithms are known since [3], up to [37], while lower bounds remain elusive, leaving the optimal query complexity still unknown.

Obtaining lower bounds is not only of fundamental interest in itself, but may also lead to interesting new techniques, since none of the existing proof strategies are applicable. New ideas provided here could prove useful to tackle other problems.

## 1.1 Contributions

We make progress towards characterizing the first-order query complexity of approximate Nash equilibria in finite-action zero-sum games, as a function of the number of actions $K$ and approximation level $\varepsilon$. Our contributions are the following:

**Lower bounds** We provide the first lower bounds on the first-order query complexity for zero-sum matrix games with bounded entries. We show that $K/2 - 1$ queries are necessary to compute an exact equilibrium (Theorem 11 in Section 4), and at least $\Omega(\log(1/\varepsilon K)/\log(K))$ queries are required for an $\varepsilon$-equilibrium if $\varepsilon \leqslant 1/K^4$ (Theorem 17 in Section 4). More than the concrete rates, we believe that the structure of our proof is of particular interest, and might be of use beyond this specific setting.

**Upper Bounds** We show that if the game matrix has entries in a known countable set, then the learner can recover the full matrix in one single first-order query (Theorem 4 in Section 3). This result can be interpreted both as an upper bound on the query complexity for matrices on countable entry sets, and as an impossibility result, precluding the use of simple predefined sets of matrices as candidate objectives for proving lower bounds. This, together with the lack of rotational invariance of the action sets, sheds light on why this setting turns out to be surprisingly resistant to proving lower bounds using well-established techniques.

## 1.2 Related Work on Lower Bounds: Lower Bounds

Below we review the literature on lower bound techniques for related problems and settings. We find that none of the existing techniques apply to our setting, for three particular reasons. They either:

– Build matrices from a set of matrices with entries lying in a finite alphabet. In Theorem 4, we show that, for game matrices with entries in a known countable set, a single first-order query suffices to infer the exact Nash equilibrium. This shows in particular that any such matrix construction cannot lead to a lower bound in the first-order model.

– Rely crucially on rotational invariance of the action set, which limits the approach to the $\ell_2$-constrained or unconstrained cases, ruling out the probability simplex.

– Assume that the players select actions in the span of the observed gradients. This span assumption is not suited when the action set is not an $\ell_2$ ball, in which case the span of the gradients has no natural embedding into the action set; most algorithms leave the span, e.g. exponential weights.

**First-order Query Complexity of Minimax Optimization** Existing lower bounds [24, 33] for minmax optimization are known for finite-dimensional Euclidean spaces where $\mathcal{X}, \mathcal{Y}$ are either Euclidean balls or the whole space. Both these cases benefit from the property that the action sets are invariant under rotation, allowing for a step-by-step reduction to lower dimensional instances, as shown in the seminal work of [29, 30]. The notable exception of [14] also does not cover the case we study: they consider the computational hardness of non-convex non-concave optimisation.

In the unconstrained case with curvature, that is, if $\mathcal{F}$ is the set of strongly convex, strongly concave functions with marginal condition numbers $\kappa_x$ and $\kappa_y$, the query complexity is settled to be $\Theta(\sqrt{\kappa_x \kappa_y} \log(1/\varepsilon))$; see [24, 43] for lower bounds and [27] and references therein for upper bounds.

For the constrained case, [33] provide lower bounds for bilinear saddle-point problems. They establish a query complexity of order $\Omega(L_f D_{\mathcal{X}} D_{\mathcal{Y}}/\varepsilon)$, where $D_{\mathcal{X}}$ and $D_{\mathcal{Y}}$ are the respective diameters of the

constraint sets $\mathcal{X}$ and $\mathcal{Y}$, and $L_f$ measures the Lipschitz regularity of $f$. However, their techniques rely crucially on rotational invariance and sequentially adapted constraint sets, and they ask the open question of whether similar lower bounds can also be shown for fixed constraint sets.

**Other Query Models**  The query complexity of minimax optimization has also been studied under different feedback models. Recall that we focus on the bilinear case, where $f : p, q \mapsto p^\intercal M q$ for some game matrix $M$. [16] study the query complexity of (approximate) Nash equilibria of the game, i.e., (approximate) saddle points of $f$, under a query model where the learner chooses single entries of the matrix to observe. They show, among other results, that in order to compute an $\varepsilon$-equilibrium in $K \times K$ zero-sum games, the number of entries queried needs to be at least $\Omega(K \log K)$ when $\varepsilon = \mathcal{O}(1/\log K)$. [22] consider a setting in which the learner observes a best reponse to their query $(p, q)$, i.e., $i^\star \in \mathrm{argmin}(Mq)_i$ and $j^\star \in \mathrm{argmin} -(M^\intercal p)_j$. In that setting, they show that to compute a $1/4$-equilibrium, at least $\Omega(\sqrt{K}/(\log K)^2)$ best-response queries are necessary. Since a first-order query brings strictly more information than either a best-response query or an entry query, these lower bounds do not have direct consequences for our query model. Both of these references prove their lower bounds by building hard matrices with entries in a known finite set (e.g., $\{0, 1\}$).

**Other Lower Bounds in Saddle Point and Equilibria Computation**  For multiplayer games, strong lower bounds on the computational complexity of exact Nash equilibria have been uncovered: PPAD-hardness for computing the Nash-equilibrium in a general game, see [7–9], or in a non-convex-concave zero-sum game [14]. Regarding query complexity, [2, 21] study the single-entry query complexity for approximate correlated equilibria and Nash equilibria of games with many players. As in other references mentioned, all these bounds are built on matrices with entries in a finite set. Due to their intrinsically combinatorial nature, these techniques are less common for numerical algorithms. One of the classical methods for sample complexity lower bounds was introduced by [31] for first-order optimization. It was successfully extended to saddle-point problems [43] and recently also to unconstrained zero-sum games by [24]. However, these techniques rely on the assumption that the next iterate is chosen in the span of the previous oracle information (cf. Definition 1 in [24] or Assumption 2.1.4 in [31]). Moreover, the game constructed by [24] for a Nesterov-style lower bound results in matrices with entries from a finite alphabet. For the same reason, no proof technique inspired by lower bounds via Rademacher random variables, as in [32], can ever work.

### 1.2.1   Other Related Work

**Upper Bounds for Finite Action Zero-Sum Games**  Early on, [3, 38] show that under best-response dynamics, the average plays converge to an equilibrium. The connection to regret bounds was established by [17], who deduce a $\mathcal{O}(\log(K)/\varepsilon^2)$ query complexity, by means of a construction akin to online-to-batch conversion. More recently, [11, 12] obtained the first fast rates of order $\mathcal{O}(c(K)/\varepsilon)$, through an ingenious learning mechanism. [37] rediscovered the Optimistic Online Mirror Descent algorithm [36], and pioneered optimistic online learning regret bounds; they yield to this the day the fastest rates of $\mathcal{O}(\log(K)/\varepsilon)$ and the simplest algorithm. The instance-dependent linear-rate upper bounds of order $\lambda(M) \ln(1/\varepsilon)$ by [19, 41] are essentially incomparable to the worst-case $\mathcal{O}(\log(K)/\varepsilon)$; they are superior when $\lambda(M) \ll K$, and vacuous in the typical case $\lambda(M) \approx K$ (see the discussion below Theorem 1).

Our focus is the quality of the inferred saddle point (this optimization perspective is sometimes called *pure exploration*). Part of the literature regards the queries as actual moves made, and consequently prioritizes other objectives. In particular, many focus on studying *uncoupled dynamics*, that is, sequences of actions that separate the observations of the individual $p$ and $q$ players without allowing communication between the players. Another recent theme of interest is the last-iterate convergence of such dynamics, see e.g., [23].

**Upper Bounds for Minmax Optimization**  The vast majority of the known upper bounds in minimax optimization (and more generally in variational inequality problems), outside of finite-action zero-sum games mentioned above, concern unconstrained settings [1, 20, 28]. The literature on constrained settings is more sparse: [4] recently proved the rate of convergence of the projected extra-gradient method [26]. [42] adapt interior methods to handle constraints. [18] propose a fast algorithm when either the action sets are strongly convex, or the objective is strongly-convex strongly-concave. Recently, non-convex-concave settings have also attracted attention, see [27] and references therein.

**Upper Bounds for General-Sum Games and Correlated Equilibria**  For multiplayer games, uncoupled dynamics do not converge (in any sense) to Nash equilibria. However, regret-based procedures can find correlated and coarse correlated equilibria [5, 39, 35]. In particular, internal regret guarantees for individual players in general-sum multiplayer games were recently improved from the generic $\mathcal{O}(1/\varepsilon^2)$ for any sequences of losses, to $O(1/\varepsilon^{4/3})$ in [40], to $O(1/\varepsilon^{6/5})$ in [6] and to $O(\log(1/\varepsilon)/\varepsilon)$ in [13, 15] (we omit the dependence on the number of actions and players). Polynomial-time methods for efficient computation of exact correlated equilibria are designed by [25, 34], using their formulation as solutions to a linear program.

## 2  Setting and Notation

**General Notation**  Given a set of numbers $A \subset \mathbb{R}$, we denote by $\mathcal{M}_K(A)$ the set of $K \times K$ matrices with entries in $A$; we mainly consider the bounded-entries class $\mathcal{M}_K([-1, 1])$. A $K \times K$ zero-sum game between a minimizing $p$-player (or row player) and a maximizing $q$-player (or column player) is represented by a matrix $M \in \mathcal{M}_K(\mathbb{R})$; we restrict our attention to square $K \times K$ games. For any pair of plays $(p, q)$, the expected loss vector of the $p$-player (resp. $q$-player) is $Mq \in \mathbb{R}^K$ (resp. $-M^\intercal p \in \mathbb{R}^K$). The suboptimality gap at $(p, q)$ is

$$g(M, p, q) = \max_{j \in [K]}(M^\intercal p)_j - \min_{i \in [K]}(Mq)_i .$$

The gap $g(M, p, q)$ is non-negative for any $M, p$ and $q$. The pair of plays $(p, q)$ is said to be an $\varepsilon$-Nash equilibrium if $g(M, p, q) \leqslant 2\varepsilon$; Nash equilibria are 0-Nash equilibria. Most games we will consider will possess a unique Nash equilibrium $p, q$, and this equilibrium will be fully supported, i.e., all components of $p$ and $q$ are positive. Fully supported equilibria in finite games are also equalizing strategies, meaning that the loss vectors are then isotropic; in finite zero-sum games, this entails that $M^\intercal p = Mq = v\mathbf{1}$, where $v$ is the game-value. We follow the notation convention to abbreviate $\min(x, v) = x \wedge v$ and to hide polylogarithmic terms by $\tilde{\Omega}$ or $\tilde{O}$.

**First-Order Query Model and Objectives**  The first-order query model is an interaction protocol between a learner and a game matrix, defined as follows. Fix a time horizon $T \in \mathbb{N}$, and a set of candidate matrices $\mathcal{M} \subseteq \mathcal{M}_K(\mathbb{R})$. Over the course of $T$ rounds, the learner sequentially picks (queries) pairs of plays $(p_t, q_t)$ and observes the expected losses $(Mq_t, -M^\intercal p_t)$, where the matrix $M \in \mathcal{M}$ is fixed and unknown to the learner.

We examine the number of interactions necessary to compute an approximate equilibrium. At the end of the $T$ rounds, the learner recommends a pair $(p, q)$. Given a fixed gap level $\varepsilon \geqslant 0$ and a set of candidate matrices $\mathcal{M}$, we say a strategy achieves query complexity $T(\varepsilon; \mathcal{M})$ if for any matrix $M \in \mathcal{M}$, the strategy outputs a pair $p, q$ such that $g(M, p, q) \leqslant 2\varepsilon$ when $T \geqslant T(\varepsilon; \mathcal{M})$. (The dependence on $\mathcal{M}$ is omitted when clear from context.) We also occasionally refer to the query complexity of recovering the full matrix, which is a different task but in the same query model. In this case, the learner recommends a full matrix and achieves query complexity $T$ if the matrix guess is correct for any true matrix $M \in \mathcal{M}$.

The topic of this paper is the study of the optimal query complexity of finding $\varepsilon$-Nash equilibria in the first-order model, for the set of matrices with bounded entries $\mathcal{M}_K([-1, 1])$.

## 3  Upper Bounds

In this section we collect upper bounds for the first-order query complexity of approximate Nash equilibria. These bounds are either well-established, trivial or exploit in a miraculously striking fashion the difference between learning saddle points for matrix games taking entries in a countable (e.g., $\mathbb{Q}$) or uncountable (e.g., $[-1, 1]$) sets of values.

### 3.1  Query Complexity over $\mathcal{M}_K([-1, 1])$: Regret and Elementary Strategies

We list some well-known results from the literature and state them in terms of first-order query complexity. The results described here are compiled in Figure 1. We use the notation $u_{n:m}$ to denote the family $(u_k)_{n \leqslant k \leqslant m}$.

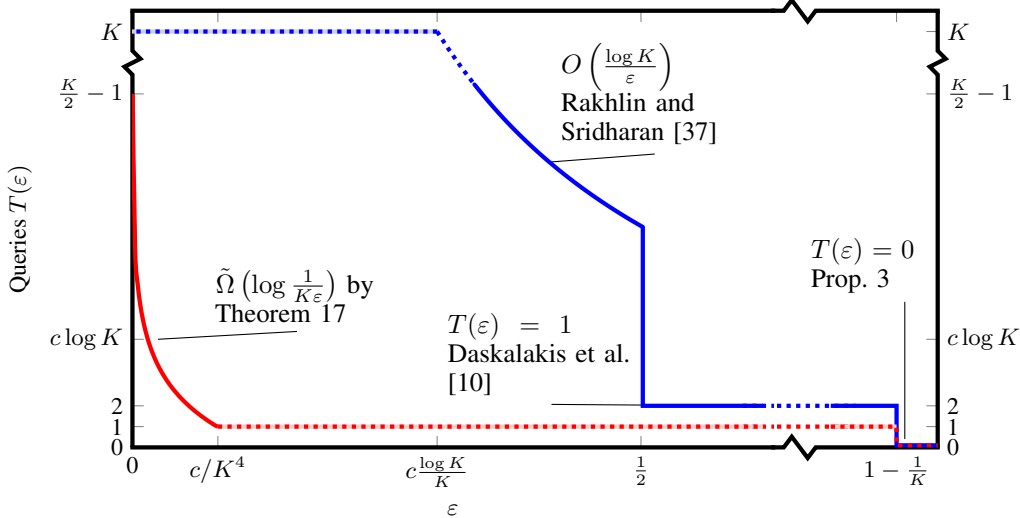

Figure 1: Upper (blue) and lower (red) bounds on the first-order query complexity of computing an $\varepsilon$-equilibrium for $K \times K$ matrix games with entries in $[-1, 1]$. See Sections 3 and 4 for details.

$O(\varepsilon^{-1} \log K)$ **Queries from Optimistic Online Learning** The current best upper bounds on the query complexity of $\varepsilon$-equilibria for zero-sum games are derived from online learning methods. As is well-known [5, Chapter 7], if we denote by $\widehat{p}_T$ (resp. $\widehat{q}_T$) the average of $p_{1:T}$ (resp. $q_{1:T}$) then

$$Tg(M, \widehat{p}_T, \widehat{q}_T) = T \max_{j \in [K]} (M^\intercal \widehat{p}_T)_j - T \min_{i \in [K]} (M \widehat{q}_T)_i$$

$$= \sum_{t=1}^{T} \langle p_t, M q_t \rangle - T \min_{i \in [K]} (M \widehat{q}_T)_i + \sum_{t=1}^{T} \langle q_t, -M^\intercal p_t \rangle - T \min_{i \in [K]} (-M^\intercal \widehat{p}_T)_j$$

$$\leqslant \sum_{t=1}^{T} \langle p_t, M q_t \rangle - \sum_{t=1}^{T} \min_{i \in [K]} (M q_t)_i + \sum_{t=1}^{T} \langle q_t, -M^\intercal p_t \rangle - \sum_{t=1}^{T} \min_{i \in [K]} (-M^\intercal p_t)_j .$$

This last term is exactly the sum of the regrets suffered by each player on their respective losses: the gap of the average plays is smaller than the sum of the average regrets over $T$ rounds. This relationship between regret and gap provides a fruitful way to upper bound the query complexity of computing equilibiria. Specifically, [37] observed that if players follow the Optimistic Mirror Descent strategy then the sum of the regrets stays smaller than $\mathcal{O}(\log K)$; see the paragraph following Proposition 6 in the mentioned reference.

**Theorem 1** (Consequence of Rakhlin and Sridharan [37]). *There exists an absolute constant $c > 0$ such that the first-order query complexity over $\mathcal{M}_K([-1, 1])$ is*

$$T(\varepsilon) \leqslant \left( c \frac{\log K}{\varepsilon} \right) \wedge K .$$

A wide stream of literature leverages the connection between regret and equilibria, and study the dynamics of pairs of learning algorithms, including the two principal flavors of optimistic online algorithms, Optimistic Mirror Descent and Optimistic Follow-the-Regularized-Leader, cf. Section 1.2.

**Instance-Dependent Rates of Convergence** Many existing bounds show rates of convergence exponential in $T$, with instance-dependent constants[1], see, e.g., [19, 41]. However, we show in Example 18 that for a zero-sum two-player bilinear game, these constants can be large, even on simple game matrices. If the constants are too large, the consequences on query complexity are vacuous, as they might require more than $K$ queries to obtain non-trivial ($\varepsilon < 1$) equilibria. Although these results give a very interesting analysis beyond the worst-case, they are beyond the scope of this work.

---

[1]More precisely: a parameter similar to a condition number of the game matrix

**Upper Bounds for Large $\varepsilon$**    We conclude this section with two elementary strategies that provide equilibria with large approximation values of $\varepsilon \geqslant 1/2$.

**Theorem 2** (Theorem. 3.1 in [10]). *For any $\varepsilon \geqslant 1/2$, the query complexity over $\mathcal{M}_K([-1, 1])$ of finding an $\varepsilon$-equilibrium is $T(\varepsilon) \leqslant 2$.*

The following proposition is a well-known observation (see, e.g., [16]) that having the learner select a pair of uniform plays always gives an approximate equilibrium.

**Proposition 3.** *For any $\varepsilon \geqslant 1 - 1/K$, the query complexity over $\mathcal{M}_K([-1, 1])$ of finding an $\varepsilon$-equilibrium is $T(\varepsilon) = 0$.*

## 3.2 Recovering the Matrix: Upper and Lower Bounds

In this section we show that the first-order query complexity for recovering the full game matrix (a task harder than computing an equilibrium) is between $\Theta(1)$, independent of the number of actions, and $\Theta(K)$, depending on the set of candidate matrices. An important consequence of these results is that several known standard lower bound techniques cannot be applied to provide lower bounds on the easier task of learning equilibria.

**Matrix Sets with Countable Alphabet**    Many existing methods rely on building hard matrices with entries in a finite alphabet: we prove this cannot lead to a lower bound.

**Theorem 4.** *Let $\mathcal{A} \subset \mathbb{R}$ be a countable set of at least two numbers. Then the first-order query complexity over $\mathcal{M}_K(\mathcal{A})$ of recovering the full matrix is $1$.*

This result exploits infinite precision in the feedback, and as such does not provide a reasonable algorithm that the players would use should they know that the matrix belongs to $\mathcal{M}_K(\mathcal{A})$. The intent of this statement is to show that any attempt at a lower bound that builds matrices with entries in a fixed countable set (and in particular with integer entries) is doomed to fail after one query.

**Remark 5.** *The proof of Theorem 4 does not provide an explicit algorithm. However, if the learner knows that the entries of the matrix are in a finite set $\mathcal{A} \subset \mathbb{Q}$, an explicit strategy can be easily described. In this case, $\mathcal{A}$ is of the form $\{a_1/r, \ldots, a_n/r\}$ with $a_i \in \mathbb{Z}$ and $r \in \mathbb{N}$. Consider a base $b = \max(a_{1:n}) - \min(a_{1:n})$ and consider a query with $p \propto (b^{-1}, b^{-2}, \ldots, b^{-K})$ (the choice for $q$ is irrelevant here). The learner can deduce the full matrix $M$ from the single observation $M^\intercal p$. Of course, this strategy has no practical interest as soon as either $\mathcal{A}$ or $K$ is moderately large, since it requires arbitrary precision in the outputs.*

**Difficulty of Recovering the Matrix Exactly**    In the first-order query model, the learner receives $2K$ numbers every round, so in order to fully determine an arbitrary $K \times K$ matrix one needs at least $K^2/(2K) = K/2$ rounds. It turns out that this bound is not tight, and we may need exactly $K$ rounds in the worst case, because of redundancies between the $2K$ numbers we observe per round:

**Theorem 6.** *The first-order query complexity over $\mathcal{M}_K([-1, 1])$ of recovering the full matrix is $K$.*

Intuitively, it is overkill to recover the matrix exactly to output an $\varepsilon$-Nash equilibrium. In the next section, when $\varepsilon \ll 1/K^K$ (in particular when $\varepsilon = 0$), we show that first-order queries do not provide useful information to find an $\varepsilon$-equilibrium faster than it takes to reveal the whole matrix.

# 4 A New Lower Bound on the Query Complexity of Approximate Equilibria

We switch perspective, attempting to make the life of any learner hard. For this, we need to think about responding to queries to keep the learner unwitting.

## 4.1 Overview and Notation

After $t$ time steps, queries $p_{1:t}$ and $q_{1:t}$ have been made and the outputs provided to the learner are $\ell_{1:t}^{(q)}$ and $\ell_{1:t}^{(p)}$. Let us denote the set of matrices with entries in $[-1, 1]$ that are consistent with the observations after $t$ time steps by

$$\mathcal{E}_t = \left\{ M \in \mathcal{M}_K([-1, 1]) \mid \forall s \in [t], \ M^\intercal p_s = -\ell_s^{(q)} \quad \text{and} \quad M q_s = \ell_s^{(p)} \right\};$$

we sometimes refer to this as the set of candidate matrices after $t$ rounds of observations. We omit the dependence on the sequence of queries $(p_s, q_s)$ and outputs $(\ell_s^{(p)}, \ell_s^{(q)})$ to reduce clutter, as it will always be clear from the context.

We say a sequence of matrices $M_{1:T}$ is adapted to the queries $p_{1:T}, q_{1:T}$, if it gives consistent outputs to the queries, i.e., if for all $s \leqslant t \leqslant T$, $M_t^\mathsf{T} p_s = M_s^\mathsf{T} p_s$ and $M_t q_s = M_s q_s$; in other words, $M_{1:T}$ is adapted if $M_{t+1} \in \mathcal{E}_t$ for all $t$.

Instead of defining directly the answers to the queries, we equivalently build a sequence of adapted matrices. Formally, at round $t + 1$, given $\mathcal{E}_t$ and $(p_{t+1}, q_{t+1})$, we select a matrix $M_{t+1} \in \mathcal{E}_t$ and output $(\ell_{t+1}^{(p)}, \ell_{t+1}^{(q)}) = (M_{t+1} q_{t+1}, -M_{t+1}^\mathsf{T} p_{t+1})$.

Let us fix some common technical notation that we use in the proofs, to measure distances in matrix space. For a matrix $M \in \mathcal{M}_K(\mathbb{R})$, and for $r, s \in [1, \infty]$, we denote the operator norm of $M$ by $\|M\|_{r,s} = \sup_{x \in \mathbb{R}^K : \|x\|_r = 1} \|Mx\|_s$. Recall that for $y, z \in \mathbb{R}^K$, we have $\|yz^\mathsf{T}\|_{r,s} = \|y\|_s \|z\|_{r'}$, where $r' \in [1, \infty]$ is such that $1/r + 1/r' = 1$. In particular $\|M\|_{1,\infty} = \max_{i,j \in [K]} |M_{i,j}|$, and $\|yz^\mathsf{T}\|_{1,\infty} = \|y\|_\infty \|z\|_\infty$. If $\mathcal{F}$ is a closed convex set subset of $\mathbb{R}^K$, we denote by $\mathrm{Proj}_\mathcal{F}(x)$ the orthogonal projection of $x$ on $\mathcal{F}$.

**Common Structure of the Proofs**    Both proofs of Theorems 11 and 17 follow the same template:

- Assume an $\varepsilon$-equilibrium is found after $T$ first-order queries.
- Find a necessary condition on this equilibrium formulated as constraints on the observed losses and the sequence of queries. Precisely, we observe that under some initial assumptions on the game matrix, the all-ones vector $\mathbf{1}$ needs to lie near the span of the outputs observed by the player, regardless of the queries and outputs.
- Ensure that this necessary condition cannot be met by any pair of mixed actions $(p, q)$ by building an appropriate adapted sequence of matrices. In our case, we ensure that the span of the losses of the $q$-player stays far from the all-ones vector $\mathbf{1}$.

This proof structure is a promising way to derive the exact query complexity of (approximate) Nash equilibria. We use it to provide the first non-trivial lower bound for this setting.

The next two sections implement this template to prove query complexity lower bounds for exact $\varepsilon = 0$ (Theorem 11) and approximate $\varepsilon > 0$ (Theorem 17) equilibria. Detailed proofs are in Appendix D. While we could have deduced the final lower bound for exact equilibria directly from the approximate equilibria case, we describe the result separately to ease exposition. Indeed, the bound for the exact case contains the main ideas but is technically simpler.

## 4.2    Proof: Exact Equilibrium Case

**Step I: Common Equilibria are in the Span of Past Queries**    We start by defining an initial set of candidate matrices with some special properties. These properties are used to ensure that the common equilibrium is fully mixed, and therefore an equalizing strategy for both players.

**Definition 7.** *A set of matrices $B_0 \subset \mathcal{M}_K([-1, 1])$ satisfies the assumptions $A(0)$ if*

- *For all $M \in B_0$, all equilibria of $M$ are fully mixed,*
- *There exists a matrix $M \in B_0$ with non-zero value.*

By Lemma 20 (App. D), any ball centered at $(1/2)I_K$ with small enough radius satisfies $A(0)$, e.g.,

$$B_0 = \mathcal{B}_{\|\cdot\|_{1,\infty}}\left(\frac{1}{2}I_K, \frac{1}{16K^2}\right) = \left\{M \in \mathcal{M}_K(\mathbb{R}) : \left|M_{i,j} - \frac{1}{2}\delta_{i=j}\right| \leqslant \frac{1}{16K^2} \quad \forall i, j \in [K]\right\}.$$

The following lemma states that after $t$ first-order queries, if at least one matrix in $B_0$ is still a candidate matrix, then any common equilibrium to all candidate matrices must lie in the span of the queries. Since equilibria of matrices in $B_0$ are fully mixed, and are thus equalizing strategies, this further implies that the all-ones vector $\mathbf{1}$ must lie in the span of the outputs to the queries.

**Lemma 8.** *Let $B_0$ be a set of matrices satisfying $A(0)$. Assume in addition that $\mathcal{E}_t \cap B_0 \neq \varnothing$ and that $\mathcal{E}_t \cap \mathcal{M}_K((-1, 1)) \neq \varnothing$. If there exists a common Nash equilibrium $(p, q)$ to all matrices in $\mathcal{E}_t$, then $p \in \mathrm{Span}(p_{1:t})$ and $q \in \mathrm{Span}(q_{1:t})$.*

**Corollary 9.** *Under the assumptions of Lemma 8, there exists a value $v \neq 0$ such that*

$$v\mathbf{1} \in \mathrm{Span}(\ell_{1:t}^{(q)}) \cap \mathrm{Span}(\ell_{1:t}^{(p)}).$$

**Step II: Sequential construction**  Given Corollary 9, to prove a query complexity lower bound, it suffices to build the answers to the queries $p_{1:t}$ and $q_{1:t}$ in a way that ensures that

- the vector $\mathbf{1}$ never belongs to the span of the observations of (say) the $q$-player,
- there is at least one remaining candidate matrix in $B_0$, i.e., $\mathcal{E}_t \cap B_0 \neq \varnothing$ .

For technical reasons (cf. the assumptions of Lemma 8), we also need to make sure that there is enough space left in $\mathcal{E}_t$. We do so by keeping a candidate matrix $M_t \in \mathcal{E}_t \cap \mathcal{M}_K((-1,1))$, away from the border of the initial set of candidate matrices (i.e., with entries stricly between $-1$ and $1$).

**Lemma 10.** *Fix a time horizon $T \leqslant K/2 - 1$. For any sequence of queries $p_{1:t}, q_{1:t}$, there exists a sequence of matrices $M_{1:T}$ in $\mathcal{M}_K((-1,1))$ adapted to $p_{1:T}, q_{1:T}$ that defines losses $\ell_{1:T}^{(q)}$ for which, for any $v \neq 0$,*

$$v\mathbf{1} \notin \operatorname{Span}(\ell_{1:T}^{(q)}) \,.$$

**Conclusion**  Combining Lemmas 8 and 10, against any learning strategy, we have built a sequence of outputs for which there is no common equilibrium to all remaining candidate matrices after $T$ rounds, as long as $T \leqslant (K-3)/2$; this proves the following theorem.

**Theorem 11.** *The first-order query complexity over $\mathcal{M}_K([-1,1])$ of finding a Nash equilibrium is*

$$T(0) \geqslant K/2 - 1 \,.$$

The next section tackles the lower bound for approximate equilibria. Its proof follows the same template as for the exact case, although the technical complexity increases.

### 4.3  Proof: Approximate Equilibria Case

**Step I: Common Equilibria Are Close to the Span of Queries**  In the following, we say a probability distribution over $[K]$ is $\delta$-supported if $p(i) \geqslant \delta$ for all $i$. We say a pair of distributions $(p, q)$ is $\delta$-supported if both $p$ and $q$ are $\delta$-supported. We start by defining a quantitative version of Definition 7 for approximate equilibria. In order to retain the property that losses at $\varepsilon$-equilibria stay close to an isotropic vector, we add a requirement on the support of the equilibria.

**Definition 12.** *A set of matrices $B_{\varepsilon,\delta} \subset \mathcal{M}_K([-1,1])$ satisfies the assumption $A(\varepsilon, \delta)$ if*

- *For all $M \in B_{\varepsilon,\delta}$, all $\varepsilon$-equilibria of $M$ are $\delta$-supported,*
- *There exists a matrix $M \in B_{\varepsilon,\delta}$ with non-zero value.*

For example, by Corollary 21, the $\|\cdot\|_{1,\infty}$-ball centered at $(1/2)I_K$ and of radius $(1/16)K^2$ satisfies this condition for any $\varepsilon \leqslant 1/(16K^2)$ and $\delta \leqslant 1/(2K)$.

The following proposition is a quantitative version of Lemma 8 for approximate equilibria, in which we show that any common approximate equilibrium to all matrices in $\mathcal{E}_t$ needs to be close to the span of the queries.

**Lemma 13.** *For $\varepsilon, \delta > 0$, let $B_{\varepsilon,\delta}$ be a set of matrices satisfying $A(\varepsilon, \delta)$, cf. Definition 12. Assume that $\mathcal{E}_t \cap B_{\varepsilon,\delta} \neq \varnothing$, and that the relative interior of $\mathcal{E}_t$ contains a ball of radius $r_t$ measured in $\|\cdot\|_{1,\infty}$-norm. If there exits a common $\varepsilon$-Nash equilibrium $(p, q)$ sto all matrices in $\mathcal{E}_t$, then*

$$\left\| p - \operatorname{Proj}_{\operatorname{Span}(p_{1:t})}(p) \right\| \leqslant \frac{2\varepsilon}{\delta r_t} \quad \text{and} \quad \left\| q - \operatorname{Proj}_{\operatorname{Span}(q_{1:t})}(q) \right\| \leqslant \frac{2\varepsilon}{\delta r_t} \,.$$

**Corollary 14.** *Under the assumptions of Lemma 13, there exists a game matrix $M \in B_{\varepsilon,\delta}$ such that the value $v \in \mathbb{R}$ of $M$ satisfies $\left\| v\mathbf{1} - \operatorname{Proj}_{\operatorname{Span}(\ell_{1:t}^{(q)})}(v\mathbf{1}) \right\| \leqslant 4\sqrt{K}\varepsilon/(\delta r_t)\,.$*

**Step II: Construction of Matrices**  We design an adapted sequence of matrices that keeps $v\mathbf{1}$ away from the span of the observed losses.

**Lemma 15.** *Let $B$ be a closed ball in $\|\cdot\|_{1,\infty}$-norm of radius $r \leqslant 1/2$ contained in $\mathcal{M}_K([-1,1])$. Fix a time horizon $T \leqslant K/2 - 1$. For any sequence of queries $(p_t, q_t)_{t \leqslant T}$, there exists a sequence of matrices $M_{1:T}$ in $B$ adapted to $(p_t, q_t)$ that defines losses $\ell_{1:T}^{(q)}$ for which, for any $v \geqslant 0$,*

$$\left\| v\mathbf{1} - \operatorname{Proj}_{\operatorname{Span}(\ell_{1:T}^{(q)})}(v\mathbf{1}) \right\|^2 \geqslant v^2 K \left( \frac{r^2}{8KT^2} \right)^{T+1} ,$$

*and such that $\mathcal{E}_t$ contains a ball of radius $r/2$ in its relative interior.*

The proof relies on a decomposition of the squared distance of $v\mathbf{1}$ to the span of observed losses at time $t+1$ through a Pythagorean identity, relating it to the span at time $t$. By carefully choosing the new matrix $M_{t+1}$ as a function of the query $p_{t+1}, q_{t+1}$, we manage to ensure that the squared distance decreases only by a constant factor.

**Remark 16.** *The radius of the ball inside $\mathcal{E}_t$ stays constant at $r/2$, even though $\mathcal{E}_t$ gets smaller as $t$ increases. This is made possible by choosing $M_t$ very close to $M_{t-1}$ (roughly at a distance of $r/(KT)$ measured in $\|\cdot\|_{1,\infty}$), effectively ensuring that $M_t$ stays far from the boundary of $\mathcal{M}_K([-1,1])$.*

**Step III: Conclusion**  We now combine Corollary 14 and Lemma 15 to obtain a lower bound on the best achievable gap after $T$ rounds, which directly translates to a query complexity lower bound.

**Theorem 17.** *In the first-order query model on $\mathcal{M}_K([-1,1])$, for any algorithm the worst-case gap after $T \leqslant (K-3)/2$ time steps is at least*

$$\varepsilon \geqslant \frac{1}{2^{10}K^4}\Big(\frac{1}{2^{11/2}K^{5/2}T}\Big)^{T+1}.$$

*Therefore, the query complexity of finding an $\varepsilon$-equilibrium for any $\varepsilon \leqslant 1/(e\,2^{11}K^4)$ is at least*

$$T(\varepsilon) \geqslant \Big(\frac{-\log(2^{11}K^4\varepsilon)}{\log(2^{11/2}K^{5/2}) + \log(-\log(2^{11}K^4\varepsilon))} - 1\Big) \wedge (K/2 - 1).$$

### 4.4   Potential Improvement and Discussion

There is still a wide disparity between upper and lower bounds on first-order query complexity. The lower bound Theorem 17 is most probably not tight, so let us discuss potential ways to improve it.

We believe the most promising approach for improvement is to find a different proxy for the gap. Our proof uses the distance to the span of the losses $\|v\mathbf{1} - \mathrm{Proj}_{\mathrm{Span}(\ell_{1:t}^{(q)})}\|^2$ for $v \geqslant 1/(2K)$, which is convenient because it is a distance, but we need to introduce a strong restriction on the set of candidate matrices $B_{\varepsilon,\delta}$ to be able to relate it to the gap, namely the restriction to a ball around $\frac{1}{2}I_K$ of radius $O(1/K^2)$. In order to make significant progress it is therefore essential to enlarge the class of candidate matrices significantly compared to $B_{\varepsilon,\delta}$, and therefore to modify the proxy for the gap.

## 5   Discussion: Future Work and Conclusion

We study the first-order query complexity of computing approximate Nash equilibria for two-player zero-sum matrix games. We review upper bounds coming from online learning, and discuss existing lower bounds for related problems including alternative query models. Taking stock, we arrive at the surprising state of affairs that for this fundamental problem no lower bounds are known. We then offer some explanation for this current state of affairs: the first-order query model is powerful enough to identify any matrix from a fixed countable set in a single query; this rules out many techniques. We then turn to lower bounds. We design an adaptive adversary that answers incoming learner queries in such a way that the remaining consistent matrices do not share a common Nash equilibrium for as long as possible. Our approach is based on a quantity serving as a "potential function": namely the distance of the all-ones vector to the span of the observations. We discuss in detail the result, future scope and limits of our proposed technique.

As can be seen in Figure 1, we are still far from matching lower and upper bounds. The most intriguing possibility for resolution would be if it were to turn out that the upper bounds (i.e. algorithms) are improvable. Current upper bounds come from online learning regret bounds, with algorithms falling in the category of uncoupled dynamics. We would love to know if these algorithm templates are in fact optimal for query complexity.

To cycle back to our motivation of computing saddle points in general, future work could attempt to extend our lower bound technique to different constraint sets, to functions possibly exhibiting curvature, and to instance-dependent rates. Another direction would be to generalize to multi-player games and investigate the query complexity of weaker solution concepts.

**Limitations and Broader Impact**  The main limit of this work is that it only partially resolves the first-order query complexity of approximate NE in finite games, calling for tighter analysis. Regarding broader impact, results are mainly theoretical and do not entail direct societal consequences.

## Acknowledgments and Disclosure of Funding

Sachs and Van Erven were supported by the Netherlands Organization for Scientific Research (NWO) under grant number VI.Vidi.192.095. Part of the research was performed while Hadiji was at the University of Amsterdam. During that time he was supported by the same grant number VI.Vidi.192.095.

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

# A   Proofs of Lemmas 8 and 10

***Proof of Lemma 8.*** Since $p, q$ is an equilibrium to at least one matrix in $B_0 \cap \mathcal{E}_t$, it is fully supported; in particular $p$ is an equalizing strategy for any $M \in \mathcal{E}_t$ and $M^\mathsf{T} p = v_M \mathbf{1}$ for some number $v_M \in \mathbb{R}$. Furthermore, the value $v_M$ is actually independent of $M \in \mathcal{E}_t$ since, $v_M = \langle M^\mathsf{T} p, q_1 \rangle = \langle p, \ell_1^{(p)} \rangle$. Therefore, for any $M, M' \in \mathcal{E}_t$, we have $(M - M')^\mathsf{T} p = \mathbf{0}$. Let us now define $\bar{p} = p - \mathrm{Proj}_{\mathrm{Span}(p_{1:t})}(p)$, and consider the direction $\bar{p} u_q^\mathsf{T} \in \mathcal{M}_K(\mathbb{R})$ for some arbitrary non-zero vector $u_q$ orthogonal to $q_1, \dots, q_t$. Fix some matrix $M \in \mathcal{E}_t \cap \mathcal{M}_K((-1, 1))$; a non-empty set by assumption. Then for $\alpha \in \mathbb{R}$ small enough, the matrix $M' = M + \alpha \bar{p} e_q^\mathsf{T}$ is still in $\mathcal{E}_t$, since $M$ is not on its border (all entries are away from $\{-1, 1\}$). Therefore,

$$\mathbf{0} = (M - M')^\mathsf{T} p = \alpha \langle p, \bar{p} \rangle e_q = \alpha \|\bar{p}\|^2 u_q \, ;$$

implying that $\|\bar{p}\| = 0$, i.e. that $p \in \mathrm{Span}(p_{1:t})$; similar reasoning shows that $q \in \mathrm{Span}(q_{1:t})$.   □

***Proof of Lemma 10.*** We build the sequence $M_{1:T}$ incrementally by moving at step $t+1$ in directions chosen as a function of the new queries $p_{t+1}$ and $q_{t+1}$, ensuring by induction that $\mathbf{1} \notin \mathrm{Span}(\ell_{1:T}^{(q)})$.

Initialize the sequence at $M_0 = (1/2) I_K$. Now for $t \geqslant 0$, let us assume that we have correctly built $M_{1:t}$, inducing losses such that $\mathbf{1} \notin \mathrm{Span}(\ell_{1:t}^{(q)})$, and let us define $M_{t+1}$. (Note that the initialization of the induction is valid with the convention that $\mathbf{1} \notin \{\mathbf{0}\} = \mathrm{Span}(\varnothing)$.)

If $p_{t+1}$ is in the span of $p_{1:t}$, then set $M_{t+1} = M_t$ and the induction holds, since the span of the losses is left unchanged. Otherwise set $M_{t+1} = M_t + \frac{\bar{p}_{t+1}}{\|\bar{p}_{t+1}\|^2} u_t^\mathsf{T}$, where $\bar{p}_{t+1} = p_{t+1} - \mathrm{Proj}_{\mathrm{Span}(p_{1:t})}(p_{t+1})$, and $u_t$ is a non-zero vector orthogonal to the vectors $q_{1:t}$, to $\ell_{1:t}^{(q)}$, to $\mathbf{1}$ and to $M_t^\mathsf{T} p_{t+1}$; the existence of such a $u_t$ is guaranteed since $2t + 2 < K$, ensuring that there is at least one dimension orthogonal to those $2t + 2$ vector. By choosing the norm of $u_t$ to be small enough, we can make sure that $M_{t+1} \in \mathcal{E}_t \cap \mathcal{M}_K((-1, 1))$. Then, $\ell_{t+1}^{(q)} = M_{t+1}^\mathsf{T} p_{t+1} = M_t^\mathsf{T} p_{t+1} + u_t$, and $\mathbf{1}$ is not in the span of $\ell_{1:(t+1)}^{(q)}$. Indeed, assume by contradiction that there exists real numbers $\alpha_{1:(t+1)}$ such that

$$\mathbf{1} = \sum_{s=1}^{t} \alpha_s \ell_s^{(q)} + \alpha_{t+1} \ell_{t+1}^{(q)} \, ,$$

then $\alpha_{t+1} > 0$ since by induction $\mathbf{1} \notin \mathrm{Span}(\ell_{1:t}^{(q)})$. Then, taking dot products with $u_t$ on both sides of the equation above, we obtain $0 = \langle u_t, \mathbf{1} \rangle = \langle u_t, \alpha_{t+1} \ell_{t+1}^{(q)} \rangle = \alpha_{t+1} \|u_t\|^2$, leading to a contradiction. Therefore $\mathbf{1} \notin \mathrm{Span}(\ell_{1:t}^{(q)})$ at all times $t \leqslant T$.   □

# B   Details for Upper bounds

## B.1   Instance-Dependent Query Complexity

The following example shows that the bounds in [19] can be vacuous for our setting. Since, as noted by the authors, Theorem 5 by [41] is equivalent to the results in [19], the following example is only with respect to the latter.

**Example 18.** *For this example, we use the notation from [19]. Consider the game matrix $M = I_K$. Observe that $M$ has a unique Nash-equilibrium, hence, by considering $p = \frac{1}{K} \mathbf{1}$ and $q = \delta_i$ it can be seen that $\delta(M)$ is at most $\sqrt{\frac{K}{K-1}} \frac{1}{K} \approx \frac{1}{K}$. The bound is defined with respect to the condition number $\kappa(M) = \sqrt{\lambda_{\max} M^\mathsf{T} M} / \delta(M)$, where $\lambda_{\max}(M)$ denotes the largest eigenvalue of $M$. Hence, for our example, $\kappa(M)$ is at least of the order of $K$, which makes a bound of $\kappa(M) \log \frac{1}{\varepsilon}$ for this specific example vacuous. Note that for other game matrices, the results give valuable insights into guarantees beyond the worst-case.*

## B.2   Constant Queries for Games from a Finite Alphabet

***Proof.*** Let $F$ be the smallest subfield of $\mathbb{R}$ that contains $\mathcal{A}$, then $F$ is countable and, $\mathbb{R}$ can be seen as an infinite-dimensional vector space over $F$. Recall that a family of real numbers $(x_1, \dots, x_n)$

(each seen as a vector over the field $F$) is linearly independent if for any $\lambda_1, \ldots, \lambda_n \in F$ we have $\lambda_1 x_1 + \cdots + \lambda_n x_n = 0$ if and only if $\lambda_1 = \cdots = \lambda_n = 0$.

We claim that if a player, say the $q$-player, queries an action such that the components $q_1, \ldots, q_K$ form a linearly independent family, then they can compute the whole matrix $M$ with just one observation. Indeed, if $M, M' \in \mathcal{M}_K(\mathcal{A})$, yield the same output after one query, then for any $i \in [K]$:

$$\sum_{i=1}^{K} (M_{i,j} - M'_{i,j}) q_i = 0.$$

Therefore, by the independence of $(q_i)_{i \in [K]}$, this implies that $M_{i,j} = M'_{i,j}$ for any $(i,j)$. In other words, no two different matrices can give the same output after one query under $q$.

We are now left to show that there exists a play $(q_1, \ldots, q_K)$ with coordinates forming a linearly independent family. We prove this using the probabilistic method. Consider a sequence of random variables $(U_1, \ldots, U_K)$ i.i.d. and uniformly distributed over $[0,1]$. Then with probability 1, the $U_i$ are independent over $F$. Indeed, we can upper bound the probability that they are dependent by a union bound and use of the tower rule as

$$\mathbb{P}\big[\exists i \in [K] \quad \text{s.t.} \quad U_i \in \mathrm{Span}_F\{U_j \mid j \neq i\}\big] \leqslant \sum_{i=1}^{K} \mathbb{P}\big[U_i \in \mathrm{Span}_F\{U_j \mid j \neq i\}\big]$$

$$= \sum_{i=1}^{K} \mathbb{E}\big[\mathbb{P}\big[U_i \in \mathrm{Span}_F\{U_j | j \neq i\} \mid \{U_j \mid j \neq i\}\big]\big] = 0.$$

The last equality holds because, for any $i \in [K]$, conditionally on $\{U_j \mid j \neq i\}$, the span of $\{U_j \mid j \neq i\}$ is a countable set, therefore the probability that $U_i$ belongs to that set is null. This concludes the proof. $\qquad\square$

## C First-Order Query Complexity of Recovering the Game Matrix

***Proof of Theorem 6.*** Clearly, we can fully reconstruct $M$ from the queries $p_t = q_t = e_t$ for $t = 1, \ldots, K$, where $e_t$ denotes the standard basis vector in direction $t$. It turns out that this is optimal.

To show this, note that each query $(p, q)$ provides us with constraints

$$p^\mathsf{T} M = a, \qquad\qquad\qquad Mq = b,$$

for some loss vectors $a$ and $b$. These may equivalently be expressed as linear constraints in the Hilbert space of matrices $A \in \mathbb{R}^{K \times K}$, with inner product $\langle A, B \rangle = \mathrm{Tr}(A^\mathsf{T} B)$:

$$\langle M^\mathsf{T}, e_i p^\mathsf{T} \rangle = a_i \qquad (i = 1, \ldots, K),$$
$$\langle M^\mathsf{T}, q e_j^\mathsf{T} \rangle = b_j \qquad (j = 1, \ldots, K).$$

Among these $2K$ constraints on $M$, there is (at least) one redundant constraint, because there always exist numbers $\lambda_1, \ldots, \lambda_K$ and $\gamma_1, \ldots, \gamma_K$, at least one of which is nonzero, such that

$$\sum_{i=1}^{K} \lambda_i e_i p^\mathsf{T} + \sum_{i=1}^{K} \gamma_j q e_j^\mathsf{T} = 0.$$

Specifically, this holds for $\lambda_i = q_i$ and $\gamma_j = -p_j$. It follows that a query $(p_t, q_t)$ in round $t$ will provide at most $2(K - t) + 1$ new constraints on top of the queries from rounds $1, \ldots, t - 1$. To see this, note that $p_t$ will have at least one constraint in common with $q_1, \ldots, q_{t-1}$ and $q_t$ will have at least one constraint in common with $p_1, \ldots, p_t$, so the total number of new constraints is at most $n_t := 2K - (t-1) - t = 2(K - t) + 1$.

We now provide the following scenario in which $M$ cannot be fully determined by strictly less than $K$ queries. In rounds $t = 1, \ldots, K - 1$, the answer to queries $(p_t, q_t)$ is always the two loss vectors $\ell_t^{(p)} = \ell_t^{(q)} = \frac{1}{2}\mathbf{1}$, which are compatible with the possibility that $M$ equals $\frac{1}{2}\mathbf{1}\mathbf{1}^\mathsf{T}$, i.e. the matrix with all entries equal to $1/2$. We will show by induction that the dimension of the null-space (i.e. the

number of unconstrained dimensions of $M$) of all constraints up to round $t$ is at least $(K-t)^2$. This is true for $t = 0$, because the domain of $M$ has $K^2$ dimensions, and, whenever it is true for $t$, then for $t + 1$ it is at least

$$(K-t)^2 - n_{t+1} = (K-t)^2 - 2(K-t-1) - 1 = (K-t)^2 - 2(K-t) + 1 = (K-t-1)^2.$$

Thus, after $K-1$ rounds, there remains at least one direction $V \in \mathbb{R}^{K \times K}$ in this null space with $V \neq 0$. This means that the learner cannot distinguish the case $M = \frac{1}{2}\mathbf{1}\mathbf{1}^\mathsf{T}$ from the case $M = \frac{1}{2}\mathbf{1}\mathbf{1}^\mathsf{T} + \frac{1}{2}V/\max_{i,j}|V_{i,j}|$, thus $K-1$ queries are not sufficient to fully determine $M$. $\qquad\square$

## D   Proofs and Technical Results for Section 4

### D.1   Approximate Equilibria

#### D.1.1   Proofs of Main Results

***Proof of Corollary 14.*** If $(p, q)$ is a common $\varepsilon$-NE to all matrices in $\mathcal{E}_t$, then we have for any game matrix $M \in \mathcal{E}_t$,

$$M^\mathsf{T}(p - \bar{p}) \in \mathrm{Span}\left(\ell_{1:t}^{(q)}\right)$$

and therefore for any $v > 0$,

$$\|v\mathbf{1} - \mathrm{Proj}_{\mathrm{Span}(\ell_{1:t}^{(q)})}(v\mathbf{1})\| \leqslant \|v\mathbf{1} - M^\mathsf{T}(p - \bar{p})\| \leqslant \|v\mathbf{1} - M^\mathsf{T}p\| + \|M^\mathsf{T}\bar{p}\| \leqslant \|v\mathbf{1} - M^\mathsf{T}p\| + \sqrt{K}\|\bar{p}\|.$$

Now as $\delta > 0$, any $M \in \mathcal{B}_{\varepsilon,\delta}$ has a unique Nash equilibrium $p^\star, q^\star$, which is fully supported. Therefore, the value of $M$ is $v = p^\mathsf{T}Mq^\star$, as $q^\star$ is a an equalizing strategy. Now, using (1) in the proof of Lemma 13, which is valid for any matrix that admits $p, q$ as an $\varepsilon$-NE,

$$\|v\mathbf{1} - M^\mathsf{T}p\| \leqslant \sqrt{K}\|v\mathbf{1} - M^\mathsf{T}p\|_\infty \leqslant \frac{\sqrt{K}\varepsilon}{\delta}.$$

$\qquad\square$

***Proof of Lemma 13.*** First note that $p$ and $q$ are $\delta$-supported, as $(p, q)$ is an $\varepsilon$-equilibrium of at least one matrix in $\mathcal{E}_t \cap B(\varepsilon, \delta)$. For $M \in \mathcal{E}_t$, pick any $j^\star \in \arg\min_{j \in [K]}(M^\mathsf{T}p)_j$. As $(p, q)$ is an $\varepsilon$-equilibrium for $M$,

$$\left(1 - q(j^\star)\right)\max_{j \in [K]}(M^\mathsf{T}p)_j + q(j^\star)\min_{j \in [K]}(M^\mathsf{T}p)_j \geqslant \sum_{i=1}^{K} q(j)(M^\mathsf{T}p)_j \geqslant \max_{j \in [K]}(M^\mathsf{T}p)_j - \varepsilon,$$

and therefore, dividing by $q(j^\star) \geqslant \delta$,

$$\max_{j \in [K]}(M^\mathsf{T}p)_j - \min_{j \in [K]}(M^\mathsf{T}p)_j \leqslant \frac{\varepsilon}{q(j^\star)} \leqslant \frac{\varepsilon}{\delta}. \tag{1}$$

Hence, as $\min_{j \in [K]}(M^\mathsf{T}p)_j \leqslant p^\mathsf{T}Mq_1 \leqslant \max_{j \in [K]}(M^\mathsf{T}p)_j$,

$$\|M^\mathsf{T}p - (p^\mathsf{T}Mq_1)\mathbf{1}\|_\infty \leqslant \frac{\varepsilon}{\delta}.$$

Since $p^\mathsf{T}Mq_1 = p^\mathsf{T}\ell_1^{(p)}$ has the same value for any $M \in \mathcal{E}_t$, we apply this inequality for any pair of matrices $M, M' \in \mathcal{E}_t$ to deduce that

$$\|(M - M')^\mathsf{T}p\|_\infty \leqslant \frac{2\varepsilon}{\delta}. \tag{2}$$

Let us now instantiate this identity with some well-chosen $M$ and $M'$. Let $u \in \mathbb{R}^K$ be a vector orthogonal to $q_1, \ldots, q_t$, such that $\|u\|_\infty = 1$ (such a $u$ exists as long as $q_1, \ldots, q_t$ do not span the whole of $\mathbb{R}^K$). Next consider $M$ and $M'$ in $\mathcal{E}_t$ such that

$$M - M' = \frac{r_t}{\|\bar{p}\|_\infty}\bar{p}u^\mathsf{T}.$$

Such $M$ and $M'$ are guaranteed to exist in $\mathcal{E}_t$ as $\|M - M'\| = r_t$. Then, applying (2), we obtain

$$\frac{2\varepsilon}{\delta} \geqslant \|(M - M')^\mathsf{T}p\|_\infty = \frac{r_t}{\|\bar{p}\|_\infty}\|\bar{p}\|_2^2\|u\|_\infty \geqslant r_t\|\bar{p}\|_2,$$

from which the claim follows. The same reasoning applies to obtain the bound on $q$. $\qquad\square$

**Proof of Lemma 15.** Denote by $\Pi_t$ the projection on the span of the observed losses $\ell_{1:t}^{(q)}$ for the $q$-player at time step $t$. We build the sequence $(M_t)$ incrementally by moving at step $t+1$ in directions chosen as a function of the new queries $p_{t+1}$ and $q_{t+1}$.

We initialize the sequence at $M_0$ the center of $B$. Now for $t \geqslant 0$, if $p_{t+1}$ is in the span of $p_{1:t}$, then set $M_{t+1} = M_t$, otherwise set

$$M_{t+1} = M_t + \frac{\bar{p}_{t+1}}{\|\bar{p}_{t+1}\|^2} u_t^\mathsf{T} \,,$$

where $\bar{p}_{t+1} = p_{t+1} - \mathrm{Proj}_{\mathrm{Span}(p_{1:t})}(p_{t+1})$, and $u_t$ is a non-zero vector orthogonal to the vectors $q_{1:t}$, to $\ell_{1:t}^{(q)}$, to $\mathbf{1}$, and to $M_t^\mathsf{T} p_{t+1}$; the existence of such a $u_t$ is guaranteed since the only condition is that it is orthogonal to $2t + 2 < K$ vectors. (Note that $u_t$ does not depend on the value of $v$.) We set the norm of $u_t$ at a later stage of the proof. Then,

$$\ell_{t+1}^{(q)} = M_{t+1}^\mathsf{T} p_{t+1} = M_t^\mathsf{T} p_{t+1} + u_t \,.$$

Then for any $v \in \mathbb{R}$, the squared distance from the vector $v\mathbf{1}$ to the space $\mathrm{Span}(\ell_{1:t+1}^{(q)})$ can be decomposed thanks to the Pythagorean equality, as the squared distance to the previous span minus the squared norm of the projection on the new orthogonal direction $\ell_{t+1}^{(q)} - \Pi_t(\ell_{t+1}^{(q)})$:

$$\|v\mathbf{1} - \Pi_{t+1}(v\mathbf{1})\|^2 = \|v\mathbf{1} - \Pi_t(v\mathbf{1})\|^2 - \underbrace{\frac{\langle v\mathbf{1}, M_t^\mathsf{T} p_{t+1} + u_t - \Pi_t(M_t^\mathsf{T} p_{t+1} + u_t)\rangle^2}{\|M_t^\mathsf{T} p_{t+1} + u_t - \Pi_t(M_t^\mathsf{T} p_{t+1} + u_t)\|^2}}_{:=D_t} \,.$$

Observe that for any vectors $a, b$, we have $\langle a - \Pi_t(a), b\rangle = \langle a - \Pi_t(a), b - \Pi_t(b)\rangle = \langle a, b - \Pi_t(b)\rangle$, as $\Pi_t$ is an orthogonal projection on a linear subspace. Using this identity, as well as the orthogonality conditions used to define $u_t$ (precisely, that $\Pi_t(u_t) = \mathbf{0}$, and that $u_t$ is orthogonal to $M_t^\mathsf{T} p_{t+1}$ and $\Pi_t(M_t^\mathsf{T} p_{t+1})$), we obtain after applying Cauchy-Schwarz,

$$
\begin{aligned}
D_t &= \frac{\langle v\mathbf{1} - \Pi_t(v\mathbf{1}), M_t^\mathsf{T} p_{t+1} + u_t\rangle^2}{\|M_t^\mathsf{T} p_{t+1} + u_t - \Pi_t(M_t^\mathsf{T} p_{t+1} + u_t)\|^2} \\
&= \frac{\langle v\mathbf{1} - \Pi_t(v\mathbf{1}), M_t^\mathsf{T} p_{t+1} + u_t\rangle^2}{\|u_t\|^2 + \|M_t^\mathsf{T} p_{t+1} - \Pi_t(M_t^\mathsf{T} p_{t+1})\|^2} \\
&= \frac{\langle v\mathbf{1} - \Pi_t(v\mathbf{1}), M_t^\mathsf{T} p_{t+1} - \Pi_t(M_t^\mathsf{T} p_{t+1})\rangle^2}{\|u_t\|^2 + \|M_t^\mathsf{T} p_{t+1} - \Pi_t(M_t^\mathsf{T} p_{t+1})\|^2} \\
&\leqslant \|v\mathbf{1} - \Pi_t(v\mathbf{1})\|^2 \frac{\|M_t^\mathsf{T} p_{t+1} - \Pi_t(M_t^\mathsf{T} p_{t+1})\|^2}{\|M_t^\mathsf{T} p_{t+1} - \Pi_t(M_t^\mathsf{T} p_{t+1})\|^2 + \|u_t\|^2} \\
&\leqslant \|v\mathbf{1} - \Pi_t(v\mathbf{1})\|^2 \frac{\|M_t^\mathsf{T} p_{t+1}\|^2}{\|M_t^\mathsf{T} p_{t+1}\|^2 + \|u_t\|^2} \,.
\end{aligned}
$$

We also used the fact that projections reduce the norm, and the function $x \mapsto x/(x+1)$ is increasing on $(0, +\infty)$ to obtain the final inequality.

Using this bound on $D_t$ to lower bound the distance to the span, we obtain

$$\|(I_K - \Pi_{t+1})v\mathbf{1}\|^2 \geqslant \|(I_K - \Pi_t)v\mathbf{1}\|^2 \left(1 - \frac{1}{1 + \|u_t\|^2/\|M_t^\mathsf{T} p_{t+1}\|^2}\right) \,. \tag{3}$$

We are now left to choose the norm of $u_t$; the objective is to make it as big as possible under the constraint that the sequence $(M_t)$ stays in $B$. We set the norm of $u_t$ to be a constant multiple of $\|M_t^\mathsf{T} p_{t+1}\|$, i.e.,

$$\|u_t\| = \sqrt{\alpha}\|M_t^\mathsf{T} p_{t+1}\| \,.$$

For $\alpha$ small enough, we can ensure that the whole sequence $(M_t)$ stays in the ball $B$, as

$$
\begin{aligned}
\|M_t - M_0\|_{1,\infty} &\leqslant \sum_{s=1}^{t} \frac{1}{\|\bar{p}_{s+1}\|^2} \|\bar{p}_{s+1} u_s^\mathsf{T}\|_{1,\infty} = \sum_{s=1}^{t} \frac{\|\bar{p}_{s+1}\|_\infty}{\|\bar{p}_{s+1}\|^2} \|u_s\|_\infty = \sum_{s=1}^{t} \frac{\|\bar{p}_{s+1}\|_\infty}{\|\bar{p}_{s+1}\|^2} \|u_s\|_\infty \\
&\leqslant \sum_{s=1}^{t} \frac{\|u_s\|}{\|\bar{p}_{s+1}\|} = \sqrt{\alpha} \sum_{t=1}^{t} \frac{\|M_t \bar{p}_{t+1}\|}{\|\bar{p}_{t+1}\|} \leqslant \sqrt{\alpha}\sqrt{K} t \,,
\end{aligned}
$$

where we used $\|M_t x\| \leqslant \|M_t\|_{2,2}\|x\| \leqslant \sqrt{K}\|x\|$, and $M_t^{\mathsf{T}}\bar{p}_{t+1} = M_t^{\mathsf{T}}p_{t+1}$ since $M_t \in \mathcal{E}_t$. Therefore by taking, $\alpha = (r/2)^2/(KT^2)$, we ensure that $M_t \in B$ as $B$ contains a ball of radius $r$ centered at $M_0$. Furthermore, this choice ensures that $\mathcal{E}_t$ contains the ball of radius $r/2$ centered at $M_t$ in its relative interior as for any matrix $U$ with norm less than $r/2$, we have $\|M_t + U - M_0\| \leqslant \|M_t - M_0\| + \|U\| \leqslant r/2$, so $M_t + U \in B \subset \mathcal{M}_K([0,1])$.

Plugging back the value of $\alpha$ into (3), we get

$$\|(I_K - \Pi_{t+1})v\mathbf{1}\|^2 \geqslant \|(I_K - \Pi_t)v\mathbf{1}\|^2 \frac{\alpha}{1+\alpha} \geqslant \|(I_K - \Pi_t)v\mathbf{1}\|^2 \frac{\alpha}{2}.$$

After $T$ time steps, this implies that for any $v \geqslant 0$,

$$\|(I_K - \Pi_T)v\mathbf{1}\|^2 \geqslant \|(I_K - \Pi_0)v\mathbf{1}\|^2\Big(\frac{\alpha}{2}\Big)^T = \|v\mathbf{1}\|^2\Big(\frac{r^2}{8KT^2}\Big)^{T+1}.$$

This is the claimed result. $\qquad\square$

***Proof of Theorem 17.*** Define $B_{\varepsilon,\delta} = \mathcal{B}_{\|\cdot\|_{1,\infty}}\big(\frac{1}{2}I_K, \frac{1}{16K^2}\big)$, which is a ball of radius $r = 1/(16K^2)$. Given an algorithm, consider the sequence of matrices $M_{1:T}$ generated by Lemma 15, applied with $B = B_{\varepsilon,\delta}$; we know in particular that $M_t \in B_{\varepsilon,\delta}$, so $B_{\varepsilon,\delta} \cap \mathcal{E}_t \neq \varnothing$.

Assume now that the algorithm outputs a common $\varepsilon$-equilibrium to all matrices in $\mathcal{E}_T$. The assumptions of Corollary 14 hold, instantiated with $B_{\varepsilon,\delta}$ and $r_t = 1/(32K^2)$. Therefore there exists $M \in B_{\varepsilon,\delta}$, with value $v$ such that

$$\|v\mathbf{1} - \mathrm{Proj}_{\mathrm{Span}(\ell_{1:t}^{(q)})}(v\mathbf{1})\| \leqslant \frac{4\sqrt{K}\varepsilon}{\delta r_t}.$$

On the other hand, by Lemma 15 also instantiated with $B = B_{\varepsilon,\delta}$, we know that

$$\|v\mathbf{1} - \mathrm{Proj}_{\mathrm{Span}(\ell_{1:T}^{(q)})}(v\mathbf{1})\| \geqslant v\sqrt{K}\Big(\frac{r^2}{8KT^2}\Big)^{(T+1)/2}.$$

This implies that

$$\varepsilon \geqslant \frac{v\delta r_t}{4}\Big(\frac{r^2}{8KT^2}\Big)^{(T+1)/2}.$$

Finally, since $v$ is the value of a matrix $M \in B_{\varepsilon,\delta}$,

$$v = \min_{p \in \Delta_K} \max_{q \in \Delta_K} p^{\mathsf{T}}Mq \geqslant \min_{p \in \Delta_K} \max_{q \in \Delta_K} \frac{p^{\mathsf{T}}q}{2} - Kr \geqslant \frac{1}{2K} - \frac{K}{16K^2} \geqslant \frac{1}{4K}.$$

We conclude by replacing the constants by their values, $\delta = 1/(2K)$, and $r_t = r/2 = 1/(32K^2)$. By Lemma 19 in Appendix D, we deduce that for $(2\varepsilon) \leqslant 1/(e\,2^{10}K^4)$,

$$T + 1 \geqslant \frac{-\log(2^{11}K^4\varepsilon)}{\log(2^{11/2}K^{5/2}) + \log(-\log(2^{11=}K^4\varepsilon))}.$$

$\qquad\square$

### D.1.2  Technical Lemmas

**Lemma 19.** *For any $a, b, x > 0$, for any $\varepsilon \leqslant a/e$,*

$$\text{if}\quad \varepsilon \geqslant a\,(1/(bx))^x, \quad \text{then}\quad x \geqslant \frac{\log(a/\varepsilon)}{\log(b\log(a/\varepsilon))}.$$

***Proof of Lemma 19.*** Assume

$$\varepsilon \geqslant a\,(1/(bx))^x = e^{-x\log(bx)}$$

then

$$(\varepsilon/a)^b \geqslant e^{-bx\log(bx)}$$

thus, applying logarithms to both sides,

$$b\log(\varepsilon/a) \geqslant -bx\log bx.$$

Reformulate as, since $a \geqslant e\varepsilon$
$$bx \log bx \geqslant b \log(a/\varepsilon) \geqslant b > 0 \,.$$
We now apply Lambert's $W_0$ function, which is increasing on its main branch $[-1/e, +\infty)$, and use a standard lower bound on $W_0$ to obtain
$$\log bx \geqslant W_0(-b\log(\varepsilon/a)) \geqslant \log(-b\log(\varepsilon/a)) - \log\log(-b\log(\varepsilon/a))$$
thus
$$bx \geqslant -b\log(\varepsilon/a)/\log(-b\log(\varepsilon/a)) \,,$$
which is the claimed bound. $\qquad\square$

**Lemma 20.** *Let $\alpha \geqslant 0$ and $s > 0$ be real numbers, let $M \in \mathcal{M}_K(\mathbb{R})$ be a matrix such that*
$$\max_{i,j\in[K]} \left| M_{i,j} - sI_K \right| \leqslant \alpha \,.$$
*Then for any $\varepsilon$-Nash equilibrium $(p,q)$ of $M$ we have for any $i,j \in [K]$*
$$\min\big(p(i),\, q(j)\big) \geqslant \frac{1}{K} - \frac{2(\alpha+\varepsilon)(K-1)}{s} \,.$$

*Proof.* Let $(p,q)$ denote an $\varepsilon$-NE of $M$. Then
$$\max_{j\in[K]} (M^\intercal p)_j - \min_{i\in[K]} (Mq)_i \leqslant 2\varepsilon \,. \tag{4}$$
Now for any $j \in [K]$, since $M$ is close to $sI_K$,
$$(M^\intercal p)_j \geqslant s\, p(j) - \alpha$$
thus, using the fact that $p$ is a probability vector,
$$\max_{j\in[K]} (M^\intercal p)_j \geqslant s \max_{j\in[K]} p(j) - \alpha \geqslant \frac{s}{K} - \alpha \,. \tag{5}$$
Similarly,
$$\min_{i\in[K]} (Mq)_i \leqslant s \min_{i\in[K]} q(i) + \alpha \tag{6}$$
Combining the equations (4), (5) and (6) above, we have
$$\frac{s}{K} - \alpha - \varepsilon \leqslant s \min_{i\in[K]} q(i) + \alpha + \varepsilon \,,$$
i.e., after rearranging,
$$\min_{i\in[K]} q(i) \geqslant \frac{1}{K} - \frac{2(\alpha+\varepsilon)}{s} \,.$$
Similarly
$$\max_{j\in[K]} p(j) \leqslant \frac{1}{s} \max_{j\in[K]} \big(M^\intercal p\big)_j + \frac{\alpha}{s} \leqslant \min_{i\in[K]} q(i) + 2\frac{\alpha+\varepsilon}{s} \leqslant \frac{1}{K} + \frac{2(\alpha+\varepsilon)}{s}$$
Therefore
$$\min_{j\in[K]} p(k) \geqslant 1 - (K-1) \max_{j\in[K]} p(j) \geqslant 1 - \frac{K-1}{K} - (K-1)\frac{2(\alpha+\varepsilon)}{s}$$
$$= \frac{1}{K} - (K-1)\frac{2(\alpha+\varepsilon)}{s} \,.$$
Completing the proof. $\qquad\square$

**Corollary 21.** *For any $\varepsilon \leqslant 1/(16K^2)$, for any game matrix $M$ such that*
$$\max_{i,j\in[K]} \left| M_{i,j} - \frac{1}{2}I_K \right| \leqslant \frac{1}{16K^2}$$
*all $\varepsilon$-NE $(p,q)$ of $M$ satisfy for all $i,j \in [K]$*
$$\min\big(p(i), q(j)\big) \geqslant \frac{1}{2K} \,.$$

**Corollary 22.** *For any game matrix $M$ such that*
$$\max_{i,j\in[K]} \left| M_{i,j} - \frac{1}{2}I_K \right| \leqslant \frac{1}{16K^2}$$
*all exact Nash equilibria $(p,q)$ are fully supported.*

