# OpenReview forum: "Towards Characterizing the First-order Query Complexity of Learning (Approximate) Nash Equilibria in Zero-sum Matrix Games"
_NeurIPS.cc/2023/Conference — NeurIPS 2023 poster_

### Official Review · Reviewer_Wvc4 · 2023-06-26

**Soundness:** 3 good
**Presentation:** 3 good
**Contribution:** 3 good
**Rating:** 7
**Confidence:** 4

**Summary:**

This paper investigates the first-order query complexity of two-player zero-sum games. More precisely, given a bi-matrix zero-sum game $(A,-A^\top)$ (with $A \in \mathbb{R}^{k\times k}$) a first-order query is a set of strategies $(x,y)$. The answer to such a query is the set of vectors $(A y, -A^\top x)$ respectively denoting the payoff vectors of the row (x) and the column (y) agent. The authors try to understand what is the necessary number of first-order queries so as to compute a min-max equilibrium of the respective zero-sum game once the matrix $A$ is unknown. The authors establish that in order to compute an exact min-max equilibrium at least $\Omega(K)$ first-order queries are needed. The latter result coincides with the first-order complexity of the naive algorithm that first $learns$ the whole matrix $A$ and then computes an exact min-max equilibrium. Then the authors show that in order to compute an $\epsilon$-approximate min-max equilibrium $\Omega(\log(1/\epsilon K))$ first-order queries are needed. The latter result approximately matches the best-known upper bound ($O(\frac{\log K}{\epsilon})$) obtained by Optimistic MWU.

**Strengths:**

I think that this is a solid work. The first-order complexity model is very well-motivated since it captures exactly the information exchange model of no-regret dynamics with full information feedback. Thus any lower-bound on the first-order complexity directly implies lower bounds on the convergence rates of such no-regret dynamics. The current state of the art convergence results of such no-regret dynamics is $O( \frac{\log K}{\epsilon})$ that is approximately matched by the provided lower bound.

I also think that the paper introduces some interesting techniques that can be of use so as to completely settle the first-order query complexity for computing min-max equilibrium.


**Weaknesses:**

The only weakness of the paper is the fact that it does not match the current upper bound. That being said I believe that this work is an important first-step.

**Questions:**

Minor: You may want to include also this references for regret-based algorithms for the computation of CCE in general-sum games.

Beyond Time-Average Convergence: Near-Optimal Uncoupled Online Learning via Clairvoyant Multiplicative Weights Update, Piliouras et al. 2022

---

> ### Author Rebuttal · Authors · 2023-08-04
>
> We thank Reviewer Wvc4 for the nice comments and for the reference, we will add it to the final version.

---

> > ### Comment · Reviewer_Wvc4 · 2023-08-11
> >
> > Thank you for response. I have carefully read the other reviews as well as your responses. I believe that your result are interesting for the *learning in games* community of Neurips and thus I have decided to keep my current score.

---

### Official Review · Reviewer_EExk · 2023-06-26

**Soundness:** 3 good
**Presentation:** 3 good
**Contribution:** 3 good
**Rating:** 6
**Confidence:** 2

**Summary:**

This paper studies the query complexity lower bound of first-order methods for finding exact and approximate Nash equilibria in two-player zero-sum games. It is first established that if the matrix entries come from a fixed countable set, one query suffices to reconstruct the full matrix and hence to compute the exact Nash equilibrium. This rules out most existing lower bound constructions as they use rational or otherwise countable matrix entries. A new approach is then used to establish an $\Omega(K)$ lower bound for finding the exact Nash and $\tilde{\Omega}(\log 1/K\epsilon)$ lower bound for finding an $\epsilon$-approximate Nash (for sufficiently small $\epsilon$), where $K$ is the number of actions.

**Strengths:**

- This paper studies a fundamental yet perhaps overlooked problem in game theory; as mentioned by the authors, it is indeed surprising that none of the established lower bounds apply to this setting. To the best of my knowledge, the results obtained in this paper are indeed the first non-trivial lower bounds for noiseless first-order methods in two-player zero-sum games.
- The paper is well-written and offers a good explanation (Theorem 4) why it is non-straightforward to adapt existing constructions to two-player zero-sum games.


**Weaknesses:**

- The current lower bound rate is still quite far from the existing upper bound in terms of the dependence on $\epsilon$.

Typos:
Line 16: Although not exactly a typo, the submodular optimization problem is never mentioned in the main text.
Line 59: "build build"
Line 519: the first term should be $\lambda_1x_1$ instead.

**Questions:**

Lines 287 and 322: a common Nash equilibrium for all matrices in $\varepsilon_t$ may not exist, right? Does the statement here mean that "suppose that a common Nash equilibrium exist"?

**Limitations:**

This work is mainly theoretical and does not entail direct societal consequences

---

> ### Author Rebuttal · Authors · 2023-08-04
>
> We thank Reviewer EExk for the helpful remarks and kind comments! We will fix the typos!
>
> **The current lower bound rate is still quite far from the existing upper bound** See our answer to 'Reviewer neqq':
> We agree that our results are quite technical and do not fully
> characterize the rates of saddle-point optimization yet. However, our
> new technique provides the first significant step towards the full
> picture, and we believe it to be of interest.
>
> **Does the statement here mean that ``suppose that a common Nash equilibrium exist"?** Yes, we are assuming existence here. This does not affect the rest of the paper in any way since we were implicitly using the correct version elsewhere. We thank the reviewer for spotting this and we will rectify the statement in the final version.

---

> > ### Comment · Reviewer_EExk · 2023-08-18
> >
> > I thank the reviewers for their response which answers my questions well. After reading the authors' rebuttal and other reviewers' comments I would be most comfortable with keeping the current score.

---

### Official Review · Reviewer_nqee · 2023-07-06

**Soundness:** 4 excellent
**Presentation:** 4 excellent
**Contribution:** 4 excellent
**Rating:** 7
**Confidence:** 3

**Summary:**

Define the gap of a two player zero sum game with payoff matrix M and mixed strategies p and q as:
gap(M,p,q):=max_j (M^Tp)_j - min_i(Mq)_i
An equilibrium is called \eps approximate if the gap is at most 2\eps. An important question is to compute the approximate Nash equilibrium by observing the gradients of the objectives in a sequential manner. Of course the number of observations / queries needs to be optimized. Rakhlin and Sridharan showed that O(\ln K/\eps^2) queries suffice for K X K matrix games. The authors here improve their bound to O(\ln K/\eps) - which is optimal - for small \eps (\eps < 1/K^4). They also show a lower bound that rules out certain algorithms for improving the upper bound to match the lower bound for all \eps.



**Strengths:**

- Learning the \eps approximate Nash equilibrium is an interesting and important open problem in its own right, not to mention several applications.
-  The techniques in the paper are interesting by themselves.

**Weaknesses:**

- The actual result is fairly technical and may not be of broad interest since the improvement is marginal (only for \eps < 1/K^4).

**Questions:**

- How robust are your techniques/bounds when the observations are noisy? Often we don't have access to precise gradients.

**Limitations:**

Yes, they have been presented clearly.

---

> ### Author Rebuttal · Authors · 2023-08-04
>
> We thank Reviewer nqee for the helpful remarks and the interesting question!
>
>
> **'the actual result is fairly technical and may not be of broad interest since the improvement is marginal'** We agree with the reviewer that our results are quite technical and do not fully characterize the rates of saddle-point optimization yet. However, our new technique provides the first significant step towards the full picture, and we believe it could be of interest on its own.
>
> **'How robust are your techniques/bounds when the observations are noisy? Often we don't have access to precise gradients.'**
>
> The reviewer asks an important and interesting question. We thought about this question before and decided to exclusively focus on the deterministic case with precise feedback for the following reasons: (1) We expect that adding enough noise would significantly alter the nature of the learning problem: statistical lower bounds would force us to cover the matrix repeatedly with queries to ensure concentration. (2) Although we agree with the reviewer that noisy feedback is of high practical relevance, we consider the exact feedback model to be a more appropriate model to study and introduce a new proof technique. We believe that whenever the deterministic problem is well understood, a very interesting next step is to combine our technique with statistical lower bounds to obtain lower bounds in the model the reviewer suggests.

---

### Official Review · Reviewer_3esb · 2023-07-14

**Soundness:** 2 fair
**Presentation:** 2 fair
**Contribution:** 3 good
**Rating:** 5
**Confidence:** 5

**Summary:**

This paper delves into the complexities associated with identifying a mixed Nash equilibrium within the context of a zero-sum game. Specifically, it scrutinizes situations where the algorithm does not directly observe the payoff matrix. Instead, it learns about it by querying its product with either a column vector or row vector. The challenge of computing Nash equilibria in this model has been extensively studied, however, prior to this study, little was known about lower boundaries.

The study's central findings include a K-2 query lower bound for calculating an exact equilibrium and a lower bound that exceeds the constant when 1/epsilon is super-polynomial in K for calculating an epsilon-approximate equilibrium. The methodology for deriving these lower bounds involves the identification of an adversary capable of answering an arbitrary series of first-order queries. This adversary ensures that the all-ones vector remains considerably distanced from the linear span of query responses for the maximum possible steps, while concurrently guaranteeing that the set of matrices congruent with query responses remains non-empty.

Early on, the paper also introduces a minor result highlighting the complexities inherent in proving lower boundaries within this model. If provided with a countable set of numbers known to encompass all the entries of the payoff matrix, a single first-order query is sufficient to reconstruct these entries and consequently, identify a mixed equilibrium. Although this result is simple, its proof capitalizes on the assumption that responses to first-order queries comprise vectors of infinite-precision real numbers. While this algorithm lacks practicality (an aspect openly acknowledged in the paper), its existence accentuates the challenge associated with establishing lower bounds in a model that facilitates such powerful queries.

**Strengths:**

n/a

**Weaknesses:**

The paper has some discernible weaknesses which impact its overall effectiveness and value. They are listed as follows:

1. Lack of Clarity: The exposition in the paper lacks clarity and coherence which makes it difficult to follow the arguments and understand the results. Better structuring and clear, concise language would greatly improve the readability and accessibility of the paper.

2. Absence of Experiments: There are no empirical evaluations or experiments provided to support the theoretical results. Experimental results are key in illustrating the practical applications and feasibility of the proposed methods. Therefore, including such results is crucial for providing a comprehensive understanding of the paper's contributions.

3. Uncertain Significance of the Results: While the results of the paper are novel, their actual importance and impact on the field are uncertain. The paper would greatly benefit from a discussion explaining the broader implications of the results, their potential applications, and how they advance the current state of knowledge in the field. Without such context, it is hard to assess the true significance of the findings.

**Questions:**

This paper leaves me in a state of indecision, and being the more positive reviewer, this doesn't appear to be a promising sign for the paper. The rather tepid scores possibly reflect my collective sentiment towards the main result, which is the lower bound. It appears to be quantitatively modest, and we're uncertain about the significance of obtaining upper/lower bounds in this exact query model that permits the use of "funny bit tricks". It's unclear how vital these aspects are in terms of contributing to the field, and this ambiguity might be affecting our overall reception of the paper.

Let me give some recommendations about improving the clarity of your theoretical paper:

1. **Abstract and Introduction**: Ensure that these sections provide a clear and concise overview of the key ideas and contributions of the paper. Avoid using overly technical terms and jargon in these sections, as many readers will try to get an understanding of your work from these parts before diving into the main content.

2. **Theoretical Concepts**: Each theoretical concept, model or algorithm that you introduce should be clearly defined and explained. When introducing a new concept, briefly review its background and the relevant literature. Also, explicitly state the assumptions you are making.

3. **Equations and Theorems**: Always explain the intuition and significance behind every equation or theorem before diving into rigorous mathematical proofs. Make sure to annotate your equations adequately and define all terms and variables used. Whenever possible, accompany your mathematical results with visualizations or intuitive examples.

This crucially misses in the main part of your results

4. **Results Section**: In the results section, don't just state your results, but also explain their implications. Make sure to clearly relate your results back to the problems and questions you outlined in the introduction.

5. **Language and Flow**: Aim for a clear, concise, and precise language. Avoid long and convoluted sentences. Make sure each section and subsection follows logically from the previous ones, helping to guide the reader through your narrative.

6. **Discussion and Conclusion**: Use these sections to clearly summarize your contributions and their implications. Discuss how your results fit into the bigger picture of your research field.

7. **Appendix**: Put additional details, proofs, and technicalities that are not essential for understanding the main ideas of your paper in the appendix. This way, you can keep the main text more readable without losing any necessary detail.

Add explanatory paragraphs after theorems



I strongly believe that the paper is not  submitted in the correct conference venue.

---

> ### Author Rebuttal · Authors · 2023-08-04
>
>
> We thank Reviewer 3esb for the remarks and comments!
> We kindly ask Reviewer 3esb to read the other reviewers' feedback regarding the concerns about the clarity of our paper. In particular, please note that all other reviewers did specifically mention the clarity and soundness of our paper as a strength. However, we are happy to clarify during the discussion phase if the reviewer can point us to any specific detail that was unclear.
>
> **Lack of experiments:**
> As our paper is theoretical, we believe that experiments should be made only if they bring value by illustrating the points made. We did not find any numerical experiments that would helpfully accompany the main message of the paper, which is our new lower bound technique.
>
> **Significance of the results:** Recent years have witnessed a rising interest in saddle-point optimization, which is at the heart of practically impactful fields (e.g., GANs, Robust Optimization, Market Design). On many aspects, the theory of saddle-point optimization parallels that of minimization. For minimization problems, we have a relatively good theoretical understanding due to (matching) upper and lower bounds (under some assumptions). However, we lack these in the saddle-point case, in particular, little is known about lower bounds. Thus, we have no confirmation that our upper bounds and the corresponding algorithms are optimal. We are making a significant (as e.g. reviewer Wvc4 noted) first step towards a lower bound.
>
> **General advice on theoretical papers and lack of intuitive explanation in main part:**
> We thank the reviewer for the comments.
> However, please note that we are familiar with this kind of advice and
> that your recommendations are also up to subjective preferences. In
> particular, we do not agree that in a rigorous mathematical proof that
> is primarily written for experts, every mathematical equation must be explained. Adding intuition after every mathematical expression might make it accessible for readers who are not familiar with rigorous mathematics, but dilute the focus on the rigorous argument. Therefore, we opt for a compromise: as you can see we added a very intuitive and high-level explanation of our proof structure at the beginning of the theoretical part, but kept the focus of the proof on pure math. Please note that this style was very accessible to all other reviewers.
> However, if the reviewer could point us to specific examples which were unclear and not intuitively accessible to the reviewer, we are happy to clarify during the rebuttal phase.
>
> **General advice on theoretical papers and undefined mathematical terms in main part:** Since the reviewer is claiming that there are undefined mathematical terms in our main part, please give more details. Specifically, we kindly ask the reviewer to point out undefined variables and mathematical terms.

---

> > ### Comment · Reviewer_3esb · 2023-08-20
> > **answer**
> >
> > Thank you for response. I have carefully read the other reviews as well as your responses. I believe that your result are interesting for the learning in games community of Neurips and thus I have decided to raise my current score.

---

### Official Review · Reviewer_VSPH · 2023-07-20

**Soundness:** 3 good
**Presentation:** 3 good
**Contribution:** 3 good
**Rating:** 5
**Confidence:** 3

**Summary:**

This paper focuses on the query complexity of learning (approximate) Nash equilibria in zero-sum matrix games. The authors provide lower bounds on the query complexity for exact and approximate equilibria. They also show that if the game matrix has entries in a known countable set, the learner can recover the full matrix in one single first-order query. The paper discusses related work on lower and upper bounds for similar problems. The authors present upper bounds for the query complexity of finding approximate equilibria and introduce two strategies for obtaining equilibria with large approximation values. They also propose a potential improvement to the lower bounds by finding a different proxy for the gap. The paper includes proofs, examples, and explanations to support the theorems and corollaries related to the query complexity of learning game matrices. Overall, the paper contributes to the understanding of the query complexity of learning Nash equilibria in matrix games and provides insights into the lower and upper bounds for this problem.

**Strengths:**

The paper presents original and valuable contributions to the understanding of the query complexity of learning Nash equilibria in matrix games. It introduces new results and insights, providing lower bounds on the query complexity for both exact and approximate equilibria, and proposing a potential improvement to the lower bounds. The quality of the paper is high, with rigorous proofs, clear examples, and a comprehensive discussion of related work. The paper is also well-structured and clearly written, making it accessible to readers. In terms of significance, the paper’s contributions can inform the development of more efficient algorithms for learning Nash equilibria in matrix games, with practical applications in various domains. Overall, the paper is a valuable addition to the existing literature in this field.

**Weaknesses:**

The paper provides valuable insights into the query complexity of learning Nash equilibria in matrix games. However, the novelty of the results is somewhat limited as they largely build upon existing literature, such as the connection between regret and equilibria.

The paper partially resolves the first-order query complexity of approximate Nash Equilibria in finite games, but further analysis is needed for tighter results. The paper’s focus is primarily on theoretical analysis and proofs, without experimental evaluation, and the results are mainly theoretical with no direct societal consequences.


**Questions:**

Please refer to weaknesses.

**Limitations:**

The author has clearly acknowledged the limitations of their work, specifically that it only partially resolves the first-order query complexity of approximate Nash Equilibria in finite games and further analysis is needed for tighter results. In terms of broader impact, the results are mainly theoretical and do not have direct societal consequences.

---

> ### Author Rebuttal · Authors · 2023-08-04
>
> We thank Reviewer VSPH for the helpful remarks and comments!
>
> **Limited novelty due to heavily building on existing results:**
> We did our best to thoroughly present the state of the problem and the related literature, and to properly credit the ideas we build upon. This, we believe, highlights the novelty of our techniques.
>
> In particular, for our main results (Section 4) we develop a new lower bound proof technique that does not build on previously known methods or results. This novel technique yields the first non-trivial lower bound for saddle-point optimization, and might be of interest on its own (which was appreciated by reviewers Wvc4 and nqee).
>
> Please also note that all our main results, e.g., Theorems 4, 6 and 17,
> are not specific to equilibrium guarantees obtained via regret bounds, and apply in a much more general context. All first-order methods must satisfy the lower bounds.
>
> **Lack of experiments:**
> We did not find any numerical experiments that would helpfully illustrate the main point of the paper, which is our new lower bound technique.

---

> > ### Comment · Reviewer_VSPH · 2023-08-11
> >
> > Thank you for your response. I have carefully evaluated your response, as well as the feedback provided by other reviewers.  After careful consideration, I have decided to maintain my current score.

---

### Decision · Program_Chairs · 2023-09-21

**Decision:**

Accept (poster)

**Comment:**

The paper addresses a fundamental open question in learning: complexity of learning 2-player zero sum games. It obtains lower bounds in the first-order model. While the specific results obtained are restricted and are far from the current upper bounds, the paper is interesting because it develops a new technique for lower bounds. We expect it to be influential in guiding future works on this subject.